# Measuring the Canopy Architecture of Young Vegetation Using the Fastrak Polhemus 3D Digitizer

**DOI:** 10.3390/s24010109

**Published:** 2023-12-25

**Authors:** Kristýna Šleglová, Jakub Brichta, Lukáš Bílek, Peter Surový

**Affiliations:** Faculty of Forestry and Wood Sciences, Czech University of Life Sciences Prague, Kamýcká 129, 16500 Prague, Czech Republic; sleglova@fld.czu.cz (K.Š.); brichtaj@fld.czu.cz (J.B.); bilek@fld.czu.cz (L.B.)

**Keywords:** 3D model, Fastrak Polhemus, tree architecture, shading, *Pinus sylvestris*

## Abstract

In the context of climate change conditions, addressing the shifting composition of forest stands and changes in traditional forest management practices are necessary. For this purpose, understanding the biomass allocation directly influenced by crown architecture is crucial. In this paper, we want to demonstrate the possibility of 3D mensuration of canopy architecture with the digitizer sensor Fastrak Polhemus and demonstrate its capability for assessing important structural information for forest purposes. Scots pine trees were chosen for this purpose, as it is the most widespread tree species in Europe, which, paradoxically, is very negatively affected by climate change. In our study, we examined young trees since the architecture of young trees influences their growth potential. In order to get the most accurate measurement of tree architecture, we evaluated the use of the Fastrak Polhemus magnetic digitizer to create a 3D model of individual trees and perform a subsequent statistical analysis of the data obtained. It was found that the stand density affects the number of branches in different orders and the heights of the trees in the process of natural regeneration. Regarding the branches, in our case, the highest number of branch orders was found in the clear-cut areas (density = 0.0), whereas the lowest branching was on-site with mature stands (density = 0.8). The results showed that the intensity of branching (assessed as the number of third-order branches) depends on the total number of branches of the tree of different branch orders but also on stand density where the tree is growing. An important finding in this study was the negative correlation between the tree branching and the tree height. The growth in height is lower when the branching expansion is higher. Similar data could be obtained with Lidar sensors. However, the occlusion due to the complexity of the tree crown would impede the information from being complete when using the magnetic digitizer. These results provide vital information for the creation of structural-functional models, which can be used to predict and estimate future tree growth and carbon fixation.

## 1. Introduction

Scots pine (*Pinus sylvestris* L.) is not only one of the primary commercial tree species in the Czech Republic but is also the most widespread tree species in almost half of the European forests [1,2,3,4]. Despite a current tendency to replace conifers with broadleaves in the Czech Republic [5], the proportion of pine is still significant, especially in places where pine stands cannot be replaced yet. In areas with a substantial proportion of pine, the Czech Republic is trying to adapt management to its ecological requirements considering climate change but also to the needs of industry [6,7]. The pine currently suffers from a significant lack of moisture; its deep tap root [8] is paradoxically unable to respond to decreasing groundwater levels. Foresters in Central Europe have already tried many ways of growing Scots pine, some of which have proven successful in terms of climate change, while others have not. At the same time, the cultivation of pine within the shelterwood system is current. In order to better understand the growth response of young Scots pine trees to different degrees of shading, a unique method of measuring with a magnetic digitizer was used in our study. Knowledge of canopy architecture is increasingly required in a variety of areas, from stand establishment to harvesting [9]. Thus, to achieve the best possible quality in a mature forest stand during harvesting, we need to understand how environmental influences affect the juvenile tree. But how can we get the most accurate architecture? One way of obtaining 3D tree models is by manual measurements—this method is, however, time-consuming both for the measurements and for the evaluation of the architecture. In addition, it is always possible to view only the 2D model but not the 3D model. The most commonly used methods for forest stand surveys are photogrammetry, Lidar, or laser scanning [10].

Photogrammetry is the science of measurement from photographs, which was originally developed for mapping and survey purposes and quickly found its application in other fields, including forestry. One of the earliest documented uses of photogrammetry in forestry was in Germany in 1920 when aerial photography was used to create detailed maps of forested areas. In the following decades, photogrammetry continued to be used in forestry for a variety of purposes, including timber inventory, forest condition assessment, and forest fire mapping. It has also been used to determine the canopy architecture of individual trees [11]. Measuring 3D models using photogrammetry is time-saving, but it is not possible to represent all parts of the canopy in the models, as shown in Figure 1. Another disadvantage of photogrammetry is that the detection of the components/pixels of the tree of interest against the background of more distant trees in photography requires sophisticated programming and can have significant mistakes.

Another method that can be used to detect 3D architecture is using Lidar. The first use of Lidar (Light Detection and Ranging) in forestry dates back to the early 1990s. Lidar must be flown either by plane or drone. From the Lidar, point clouds are created that provide a direct 3D view of the surface. A specialized method is needed to extract the canopy architecture from the 3D point cloud to synthesize their qualification into useful measures of tree features. The dataset combined with algorithms representing the forest canopy resulted in a fine-scale architectural model called L-Architect (Lidar data to tree architecture) [11]. L-Architecture was proposed as a practical way to synthesize and quantify the spatial distribution of tree components from Lidar point clouds. L-Architecture uses geometrically registered Lidar scans of individual trees to reconstruct the geometric and topological structure of the tree. In our case, as we want to achieve high accuracy, this method is not suitable for our purposes (Figure 2), due to the fact that not all parts of the crown are visible.

The above-mentioned methods are more used to determine the condition of the whole forest but not to determine the trees’ architecture. This is because these methods have a large error in displaying 3D models of tree species, as they are not able to display all parts of the canopy [12]. A method for measuring three-dimensional tree architecture that works at the branch level and simultaneously describes plant topology (branching), plant geometry, and branch morphology combines a 3D digitizer [13] (Figure 3) coupled with DiplAmi software (version1.0) to control the digitizer and collect data [14]. The 3D digitizing method started to be used in 1973 when an articulated arm measuring rotation angles was used for the first 3D digitizing [15]. However, smaller plants rather than mature trees were examined using 3D digitizing equipment. Digitization of smaller plants has been addressed in several different studies, with the general conclusion being that 3D models have the potential to demonstrate differences in structure and development between individuals and under different environmental conditions [16,17,18].

The digitizer Fastrak Polhemus, due to its electromagnetic tracking (EMT), is used in various fields, not only in forestry. One of the most well-known fields is medicine, where this technology enables precise navigation deep within the human body by providing real-time data on the position and orientation of the tracked instruments used by the clinician. In every field, magnetic navigation errors occur due to distortion in the operating environment. Deformers are metal objects that can be static or move relative to the tracking system transmitter. It is not possible to completely avoid the action of a metallic part in the magnetic field, but it is possible to limit it with the use of multiple coils at once. In fields such as medicine, this option of dealing with metal parts is possible [19]. In our case, where the coils are replaced by a stylus, using multiple styluses at once is not an option. So, we must always be careful of the metal parts. Using a magnetic digitizer, we image all parts of the crowns, even those overlapping in Figure 4. This method is the most accurate but the most time-consuming. This method makes it possible to measure the 3D point coordinates of selected points in space. This method is most often used for small trees. When we measure trees taller than 4 m, we have to divide the tree into several parts and use reference points to reassemble which will be discussed in the methodology section.

This method has already been used, for example, for the architectural description of a 20-year-old, 7 m tall walnut tree, and the visual comparison between the photograph of the tree and the image synthesized from the digitization was satisfactory [20]. Mutke et al. [14] investigated the correlation of topological and geometric variables in individual trees. The correlation was confirmed only with the parameters of the parent shoot that had formed in the previous year. However, smaller plants rather than mature trees were investigated using 3D digitizing equipment. Digitization of smaller plants has been addressed by several studies, with the general conclusion that 3D models can demonstrate differences in structure and development between individuals and under different environmental conditions [16,17,18]. Water and nutrient content is among other factors that influence the growth and architecture of woody plants. Most published work on this topic has confirmed the hypothesis that transpiration is influenced by both tree morphology and environmental factors [21,22,23]. Transpiration is higher with the increase in tree height and diameter [24]. Most models of plant growth and architectural development are generally based on open modular structures and their endogenous dynamics: germination, growth, senescence, typology, and geometry [25,26,27]. Functional-structural tree models can approach crown development through shoot-level processes using a framework of various interacting sub-modules that consider both branch position in the crown and light environment as relative factors for shoot morphology [28,29,30]. Crown density is an important parameter related to canopy architecture [31]. Our study describes the measurement of architecture using a 3D digitizer in order to evaluate the morphology of individuals under different light conditions according to the actual measured data and to assess the effect of protective timber management on the crown morphology of natural regenerating Scots trees.

Crown architecture and its formation over time affect most management objectives in Scots pine, such as stem formation, the presence of knots, stem cleanliness, the lack of presence of defects, scars, or dead branches, the presence of branch scars, etc. Crown architecture, the number of branches, the number of orders, and other morphological parameters have not yet been studied, mainly due to the lack of technologies capable of providing reliable data [32]. Historically, many studies have been conducted to study the architecture of trees (specifically pine). The main focus of these studies was tree biomechanics [32,33,34], branch mechanics [32], root mechanics [28], tree pruning [29], and more recently, studies on growth and dynamics at the architectural level [10]. Our study relates to the latter in that it provides an example of how to support the technical limitations of growth studies by deploying a 3D magnetic digitizer as an alternative to Lidar scanning. We seek to demonstrate the findings on growth dynamics (branching intensities) and how current technology can easily achieve this.

Based on the facts listed above, the hypotheses of this study were as follows: (1) stand density affects the canopy branching of regeneration trees, (2) stand density affects the height of regeneration trees, and (3) canopy branching is related to the height of regeneration trees, and we can provide data to support this hypothesis using 3D magnetic digitizer technology.

## 2. Materials and Methods

### 2.1. Study Area

The subject of the study was a stand of Scots pine near the town of Doksy, on the property of Vojenské lesy a statky, s. p., Mimoň division (50°33.77548′ N, 14°43.49143′ E) (Figure 5). Almost 100% of the forest stand consists of naturally grown Scots pine. The terrain is flat at 300 m above sea level. The study areas are located in natural pine habitats, and the subsoil is sandstone with the dominant soil type arenaceous podzol [34]. The average annual air temperature is 7.3 °C, and the average maximum temperature is 31.5 °C. The average annual precipitation is 635 mm [35]. Growing seasons last approximately 162 days [36]. The study territory typically has warm, dry summers and cool, dry winters with a narrow annual temperature range (Cfb) according to the Köppen climate classification [37]. The study sites naturally host Scots pine Pinetum oligotrophic with a sparse cover of *Vaccinium myrtillus* L. and *Vaccinium vitis-idaea* L. in the herbaceous layer [38]. Natural regeneration is meant here as an alternative for seeding or planting. Thus, it means trees are from natural sources even though the soil is managed.

### 2.2. Data Collection

In 2016, an experimental research plot was established to monitor the success of the natural regeneration of Scots pine with different densities of trees per stand and different soil preparation [35]. Part of the design of the Brichta et al. [35] experiment was also utilized in this paper. Plots of interest for data collection were delineated for parts of the stand without soil preparation. Eight trees of natural regeneration were taken from each plot with different mature stand clear-cut areas—no cover shelter, stand density 0.4–40% of the original cover, stand density 0.6–60% of the original cover, and stand density 0.8–80% of the original cover (Figure 6). The trees present in the natural regeneration were selected using random stratified sampling for the laboratory study from two subplots precisely in the middle of the width of the plots with different mature stand densities. This configuration was chosen to minimize the influence from one side or the other by a different variant of stand density or by a cardinal direction effect. Thus, 32 natural regeneration trees, including their entire root systems, were sampled from a total of eight subplots in four plots with different stand densities (Figure 6). The age of all sampled trees was 5 years old.

The collected samples were analyzed with the magnetic digitizer Fastrak Polhemus (Company: NEUROSPEC AG City and country: Stans, Switzerland Version of software: 1.0) [13] (Figure 3) to produce accurate 3D models of these trees. One might consider magnetic digitizer to be the most accurate method to obtain the entire tree architecture, especially in cases where some of its parts overlap each other. The digitizer uses a stylus (Figure 7) to measure the position of the points (with an accuracy of 0.07 cm) and the given branching angles (for X, Y, and Z coordinates with an accuracy of 0.15°). The center of the Cartesian coordinate system lies in the center of the magnetic field, where the coordinates X, Y, and Z are equal to zero. During the measurement, the tree must be in the magnetic field generated by the generator (Figure 8) to obtain an accurate and complete crown architecture. The first point is the point at the base of the trunk, and the next point is the point where the branches start (we get a segment or part of the trunk) and continue to the top of the tree. Next, the first-order branch is measured the same way as the trunk, and then the entire branch with the other orders is measured (Figure 9). In this study, the measurement of first-, second-, and third-order branches was the main goal (Figure 10).

The stylus (Figure 7) works on the principle of magnetic induction. If this tip is placed in a magnetic field, after its activation, a voltage is induced according to Faraday’s law, as shown in Equation (1). The touch position is then calculated using this signal [39]:ε = −dΦB/dt(1)
where ε is the electromotive force (emf), and ΦB is the magnetic flux.

The Faraday equation states that a time variation of the magnetic field always accompanies a spatial variation in a non-conservative electric field, and vice versa.

The generator of the electromagnetic field (Figure 8) produces a magnetic field constantly, and the moment we press on the stylus, the magnetic field is disturbed. We disrupt it at a certain angle and at a certain place. The coordinates that we entered with the stylus are the coordinates of the location of the disturbed magnetic platform. Since we are working with a magnetic field that is constantly in motion, there must be no metal in our measurement area. It would disturb the correct magnetic field waves, and we would have entered the wrong coordinates for the disturbance.

The first magnetic tracking system was developed by Polhemus Navigation Systems [34], which standardized the tracking system topology. An overview of how a typical EMT (electromagnetic tracking) system operates is shown in Figure 11.

### 2.3. Data Analysis

In total, all 32 pine trees were analyzed (Figure 12) and Appendix A. For better visualization, 3D images of natural regeneration trees were created [12]. Data were processed in Fastrak Digitizer software (version 1.0) [13]. 

To calculate the branch lengths and the sum of the branch lengths of all three orders, the sum of the lengths of individual branch segments was used. The length of each segment was determined using the following formula:(2)$$L=\sum_{i=1}^{n}\;\sqrt{{\left(x_{i-s_{1}}-x_{i-s_{2}}\right)}^{2}+{\left(y_{i-s_{1}}-y_{i-s_{2}}\right)}^{2}+{\left(z_{i-s_{1}}-z_{i-s_{2}}\right)}^{2}}$$
where L—branch length, n—number of branch segments, i—order number, [X_i−_s_1_, Y_i−_s_1_, Z_i−_s_1_]—coordinates of the segment start point, [X_i−_s_2_, Y_i−_s_2_, Z_i−_s_2_]—coordinates of the segment final point.

Statistical evaluation was performed in R software (version 4.3.1.) [40]. The boxplot function was used for the visual analysis, and a generalized linear model with a Poisson probability distribution (the first variable, in this case, was stand density, and the second variable was total wood biomass) was used for the regression analysis of branching intensity dependence (number of the first-, second-, and third-order branches). The linear model with Gaussian distribution was chosen to analyze the dependence of individual height based on stand density.

## 3. Results

### 3.1. Dependence of Branching Regarding Stand Density

The most frequent occurrence of the first-, second-, and third-order branches was found in the clear-cut areas, while the lowest number of the first-, second-, and third-order branches was found in the areas with a stand density of 0.8 (80%). The first order was most frequent in the clear-cut area, while in the plots with a density of 0.4 (40%) and 0.6 (60%), there was no statistically significant difference in the number of occurrences of first-order branches (Figure 13A). The lowest abundance of the first-order branches was evident on a stand density area of 0.8. Similar results were also obtained for the abundance of the second-order branches, and statistically significant differences were observed between the clear-cut area and stand density area of 0.4. Again, the stand density of 0.8 area had the lowest occurrence of the second-order branches (Figure 13B). The third-order branches were, again, most abundant in the clear-cut area and minimal in the stand density of 0.4 area. There were no third-order branches in plots with 0.6 and 0.8 densities (Figure 13C).

Our analysis focused on the effect of stand density on the abundance of each branch order. In the table below, we present the coefficient of the generalized linear model of the relationship between stand density and the abundance of branching order (Table 1). According to the regression coefficient for each linear model (for orders 1, 2, and 3), Table 1 shows that the stand density has a statistically significant effect on the number of individual branch orders. In the case of the first-order branches, the highest number of branches is evident in the clear-cut area, with an average value of 15. The lowest occurrence is on the stand density area of 0.8. In the case of the second order, the highest number of branches is also in the clear-cut area, with an average value of 14. The lowest occurrence of the second-order branches is in the stand density area of 0.8 with an average value of 1, while in the case of the third-order branches, the highest frequency is again in the clear-cut area (Table 1). Regarding the stand density of 0.4, the third-order branches rarely occur and are completely absent in the stand density of 0.6 and 0.8. The result shows that as stand density increases, the frequency of each order of branches of natural regeneration individuals decreases.

### 3.2. Dependence of Height of Natural Regeneration Individuals on Stand Density

The tallest pine tree was found in the clear-cut area, with an average height of 90 cm, while the smallest pine trees were found in the stand density of 0.8 (15 cm) area. Pines in the stand density of 0.6 (35 cm) area were larger on average than those in the stand density of 0.4 (30 cm) area (Table 2, Figure 14). Based on the values presented, we can confirm the hypothesis that stand density has a statistically significant effect on the height of natural regeneration individuals.

In the Table 2, we examined the relationship between tree height and the stand density and the density of the parent stand. Below, you will find the interpretation of the individual columns.

The intercept value is 2.73680. We found that this value is statistically significant (*p*-value < 2 × 10^−16^ ***), which indicates that we can expect a significantly higher height of individuals even with a low density of the mother stand. The coefficient for “Stand density level” is −1.6223. A negative sign indicates that the expected tree height decreases with increasing stand density. This coefficient is again statistically significant (*p*-value 8.19 × 10^−15^ ***), which confirms that the density of the stand has a significant effect on the tree height. This information is key to understanding ecosystem dynamics and may have important implications for forestry and ecological strategies.

### 3.3. Dependence of the Height of Natural Regeneration Individuals on the Biomass of Orders

Our analysis focuses on the relationship between tree height and the total branch biomass of each order. Below, we present the interpretation of the individual columns (Table 3). The dependence of the height of a tree on the total biomass invested in the branches with respect to their order was evaluated. The model’s hypothesis that the natural regeneration of Scots pine increases their height with the same intensity as they increase their branches, i.e., that Scots pine height positively correlates with the number of branches in each order (biomass), was not confirmed (Table 3). In contrast, the statistical model shows that the height of natural regeneration is negatively correlated to the amount of biomass generated from the third-order branches; this finding can also be interpreted to mean that the more biomass a tree stores in the third-order branches, the less height it will attain.

### 3.4. Evaluation of the Time and Complexity of Measurements

Every measurement method has advantages and disadvantages. Fastrak digitizer measurements are time-consuming to collect data in the field, but after the data are processed, a field campaign is no longer required. Since the magnetic digitizer needs a power supply for the measurement, it usually has a rule of thumb that a 1 m tall tree takes about 45 min to be measured. Of course, other factors must be taken into account, for example, canopy branching. The data collected can be used to create structural models, which can help to predict the growth of a given tree and evaluate the environmental conditions. For the structural models, the GroIMP software (version 2.0.1)can be used, in which accurate 3D models are necessary in order to simulate growth as accurately as possible, for example. Three-dimensional models measured with Fastrak can therefore be used in GroIMP for growth simulations directly from the measured data without further modifications [41].

## 4. Discussion

### 4.1. Dependence of Branching on Stand Density

Lidar cannot accurately detect branch branching because Lidar cannot measure overlapping branches or branches inside the crown. However, a model canopy structure can be created using Lidar. A method has been developed that overlays a three-dimensional matrix over the forest stand. The [42,43,44] magnetic digitizer can be used to determine the model canopy structure, but to understand the exact strategy of individual trees, one needs to know the canopy branching directly. Consistent with our first hypothesis, the number of branch orders was higher in the clear-cut area, while the lowest number was in the areas of 0.8. Differences in the stand density areas of 0.4 and 0.6 were apparent but not statistically significant. The results of our study indicate that if pines grow in a clear-cut area, they will be more branched. While this helps to make the tree more stable, branched crowns may be more susceptible to potential fires [39], especially considering the climatic fluctuations in recent years [45,46,47,48,49]. Models of tree crown architecture in temperate forests are limited to a few studies [50], making it difficult to compare our work’s specific methods with similar models. However, for a tree species to survive in a competitive and restrictive environment, its architecture must provide so-called “functional advantages” [51]. For example, when silver birch (*Betula pendula* Roth.) is under severe competitive stress, it invests more in height than in diameter [44]. To a lesser extent, this is also true for Scots pine at young ages [52].

In our study, the crown development was slower in stands with higher stand density than in those with lower stand density. Another study also confirmed, for example, the tendency of Scots pine to develop its crown toward accessible light [53]. For now, however, the question remains how the crown width and the number of orders of existing seedlings will respond to further release of the parent stand. For example, Riikonen et al. [54] showed that Scots pine seedlings respond very well to a reduction in stand density just by increasing crown width. The light plasticity of the crowns of some tree species, i.e., the tendency to achieve a very characteristic crown shape, can then be strongly influenced by the inherited development of the tree [54]. However, our study is consistent with the proposition that crown architecture is crucial for light interception and distribution to each specific photosynthetic unit of the crown. Tree crown architecture can be represented by “models” that delineate basic growth strategies that define successive architectural phases [55], which is also supported by our measurements using the Fastrak Polhemus magnetic digitizer [13].

### 4.2. Dependence of Height of Natural Regeneration Individuals on Stand Density

As the magnetic digitizer provides accurate 3D models, it is possible to infer about the dependence of the tree height and the stand density. There are many studies that determine the height of vegetation using Lidar data. When measuring with a magnetic digitizer, we do not have to deal with any optimum size as it is based on the magnetic wavelength. The maximum tree height came out with the highest accuracy of 85–90% [56]. However, the accuracy of the magnetic digitizer is 99%, with only a 0.7 mm deviation from the actual tree height.

Consistent with our second hypothesis, the highest pine trees were found in the clear-cut areas, whereas the smallest were found in the areas with a stand density equal to 0.8. No statistically significant differences were found in the stand density areas of 0.4 and 0.6. This result is also confirmed by the study of Brichta et al. [35] that the highest density of pines is in the meadow, where there are also the tallest pines, followed by stand density of 0.4 and stand density of 0.6 and 0.8. Earlier literature also supports the above conclusions, namely that better growth of young pines is evident in areas with lower stand densities [57]. Hence, stand density is also characterized here as a significant indicator directly related to the crown architecture [31]. Another possible negative effect on tree heights in the shelterwood group was the presence of bilberry (*Vaccinium myrtillus* L.) [58,59]. In our plots, bilberry was abundant in the 0.8 stand density level, which also contained the lowest tree of natural pine regeneration. However, data from the study by Brichta et al. [35] suggest a contrary—positive—effect of the bilberry competition on the height of natural regeneration trees. Yet, our study showed that the higher the stand density, the smaller the heights of the trees, i.e., except for the stand density of 0.6, where on average, the trees were taller than in the area with the stand density of 0.4.

The earlier cited exception to this rule could be due to the specific microhabitat conditions at the sampling site. Thus, given the high competition at the herbaceous vegetation layer, the most critical mechanism seems to be primarily illumination, which influences shelterwood height, and, secondly, the overall architecture of woody plants [60,61]. Only tree growth was considered in our study, but other works have also examined various factors that affect the growth. For example, orthotropy is most commonly associated with conditions of high light intensity, whereas plagiotropy is related to shade conditions. Both orthotropy and plagiotropy are related to certain limiting factors and are viewed as mechanisms that allow synchronization of growth with occasional favorable conditions [62]. However, even the mere consideration of an architectural model is likely insufficient to explain all the links between the ecological status of a species and its architecture. According to Horn [60], architectural models do not directly reflect the adaptive geometry of adult plants. Several species sharing the same architectural model may take on very different shapes. The species model and strategy are only minimally related because the tree shape is flexible [63]. Our results compare favorably with those of another pioneer tree species, namely silver birch (*Betula pendula* Roth.). A study dealing with birch confirmed that birch facing intense light competition is five times smaller than birch with no competition for light [50]. Thus, the latter result also does not contradict our conclusions that natural regeneration pine trees showed the lowest heights only in the area with a stand density of 0.8.

### 4.3. Dependence of the Height of Natural Regeneration Individuals on the Biomass of Orders

Regarding our third hypothesis, the tree height is correlated only with the first- and second-order branches but not with the third-order branches. However, given the paucity of similar studies, we can relevantly discuss, in particular, why the third order did not affect the height of natural regeneration. Stevens and Perkins [64] mention a correlation between tree height and relative size in this context, but they also question the taxonomic magnitude of this effect. Our data present that the occurrence of third-order branches tended to decrease with increasing stand density. Young pines in open areas invest biomass both in lateral branches and in height growth. Our result further confirms the observation that crown branching is not desirable if the goal is to obtain wood with high quality, which was concluded by researchers investigating the effect of pruning on the tree tissue in the pine forests of Central Europe in a research plot established in 1951 [65]. The results of this work can be compared, for example, with Šebeň et al. [66], who focused on Norway spruce (*Picea abies* L. Karst), where a higher ratio between height increments and lateral branch increments was found exclusively in dominant trees than in non-dominant trees. In this case, height increments were twice as high as lateral increments. In suppressed trees, the ratio of height increments to branch increments was around 1 [66]. It should be added that branching is not very desirable in forestry [67,68,69,70,71]. The latter conclusion implies that in the case of shelterwood management, the regeneration of Scots pine can only be sufficiently rapid, and the quality of the individuals can only be ensured when the stand is considerably thinned.

## 5. Conclusions

This study verified the feasibility of measuring tree architecture using the Fastrak Polhemus magnetic digitizer. It was shown that we can measure even the smallest parts of the tree canopy with this instrument. This is especially important for young individuals, where every small detail is important for understanding their growth strategy. Their measurement is nowadays more and more desirable, especially because woody plants are most productive (in terms of carbon fixation) at a young age, and their formulation is reflected in the future. The measurement of young trees is rare, although important for the future of our forests. To understand the growth strategy of trees as much as possible, we need to know their architecture. It is by using the Fastrak Polhemus magnetic digitizer that we obtain accurate architecture. Currently, the most used method is Lidar; we only compared this method with the help of the literature because using Lidar did not seem practical given the environment, mainly because its use is not effective in young stands. However, Lidar would be more appropriate if we needed to measure mature trees in the shortest possible time. As mentioned in the above studies, Lidar cannot image all tree stand parts. However, the Fastrak Polhemus magnetic digitizer eliminates this shortcoming because it measures the inner part of the canopy, even where the branches overlap. It is a unique measurement method that, although more time-consuming in the field, has no time requirements for processing results outside the field, unlike other methods. The measurement technique used appears to be suitable for similar analytical studies and, in addition to providing non-destructive measurements, provides clear data in 3D models and is also time-efficient compared to other destructive measurements. We can continue to use these 3D data models for growth models displayed in simulation software like GroIMP to understand tree growth strategies better [41,71,72,73,74]. In this study, data on the architecture of the crown and individual branches were obtained using the aforementioned technology to assess the condition and development of Scots pine trees at different stand densities. Based on the analysis of the measured data, it can be concluded that the five-year-old trees of natural regeneration of Scots pine in the clear-cut area are the tallest and have the highest branching density, where the third-order branches are also present. Therefore, it can be concluded that the open, clear-cut area provides the most suitable conditions for the growth of young pines. However, it is interesting to note that the number of third-order branches is not positively correlated with height. Young pines in open areas invest biomass both in lateral branches and in height growth. The present study confirms the importance of sufficient solar radiation for the density and growth of the natural regeneration of Scots pine trees. Because of the limited number of samples evaluated and the lack of similar studies, we recommend that follow-up research be conducted, not only in natural pine habitat conditions.

Our main aim was predominantly to understand growth in general, not an ecological aim, but rather an industrial aim. This study is possibly one of the first research studies of this kind in Central Europe, especially with the use of the Fastrak Polhemus magnetic digitizer. For future studies, an increase in the amount of data is necessary.

## Figures and Tables

**Figure 1 sensors-24-00109-f001:**
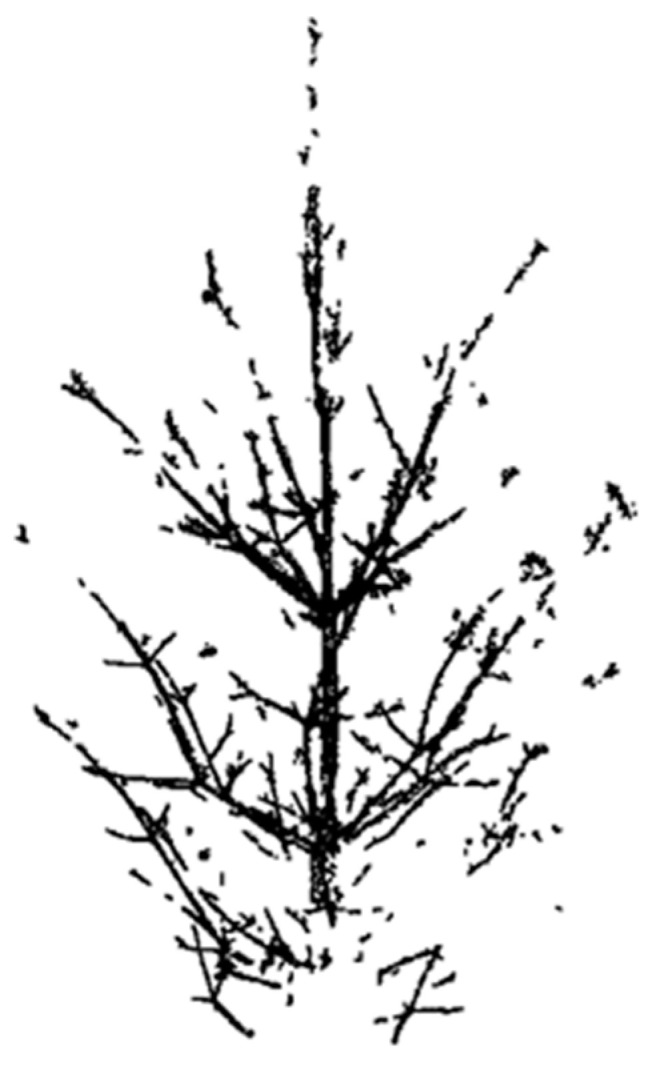
Photogrammetric 3D model of a Scots pine tree in a regeneration stand.

**Figure 2 sensors-24-00109-f002:**
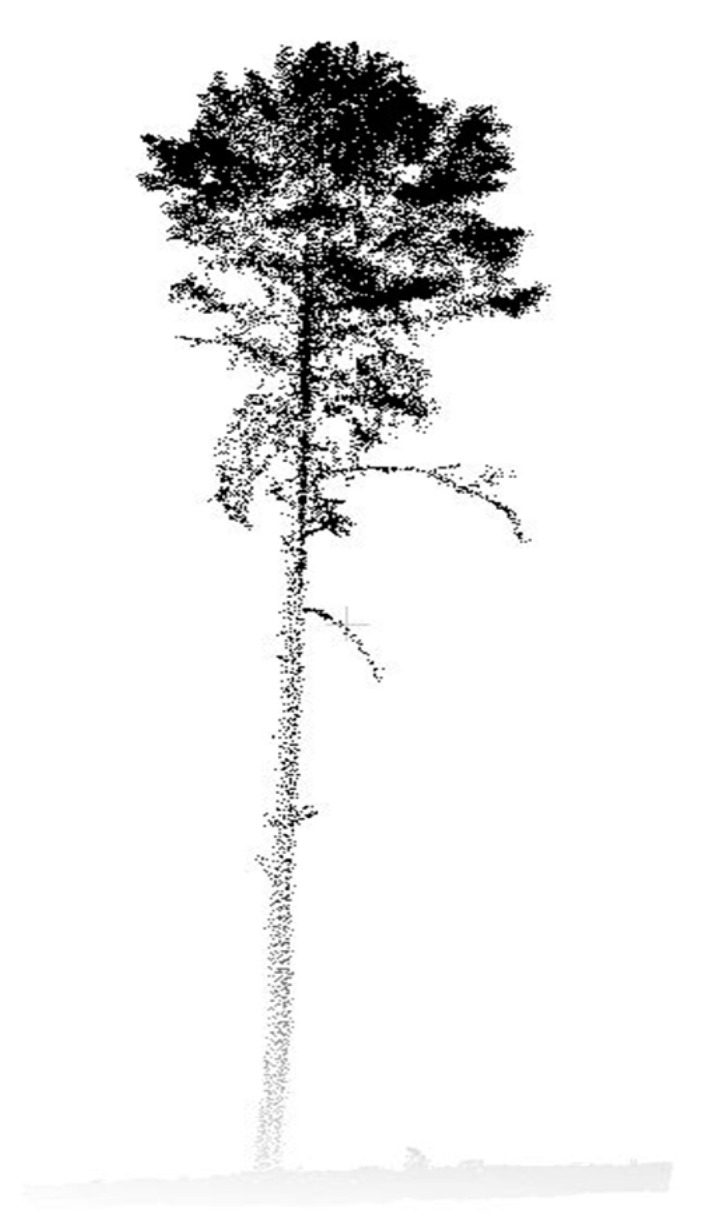
Lidar 3D model showing a stand of Scots pine tree.

**Figure 3 sensors-24-00109-f003:**
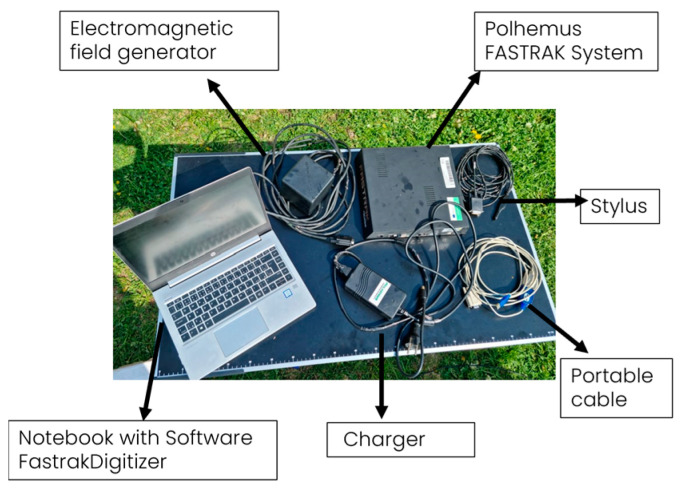
Three-dimensional magnetic digitizer Fastrak Polhemus.

**Figure 4 sensors-24-00109-f004:**
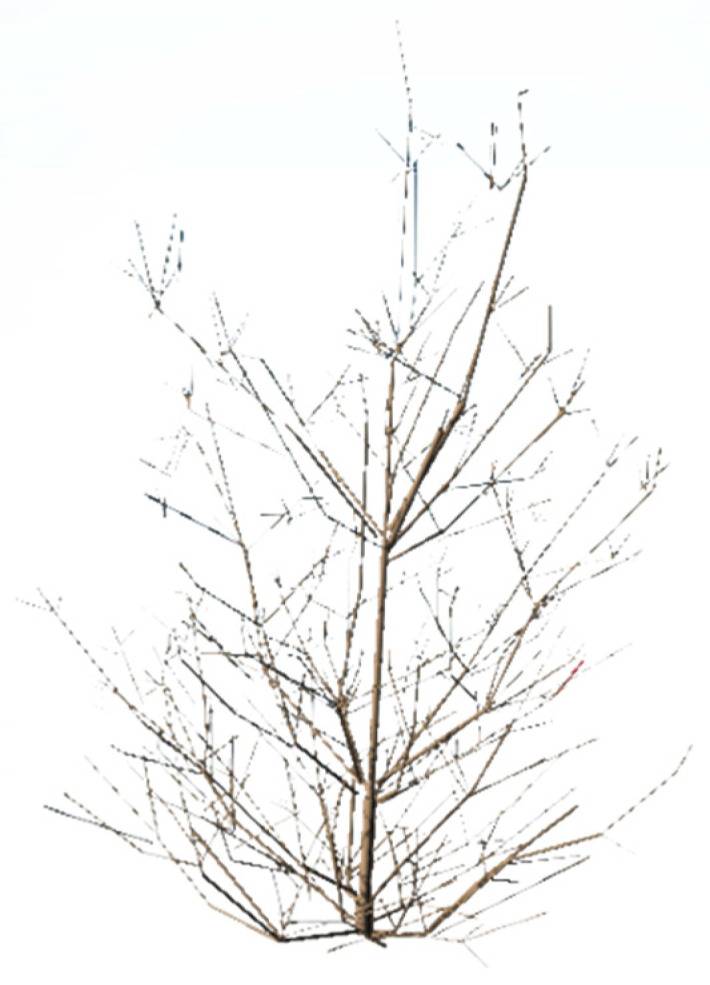
Three-dimensional model of a Scots pine tree in regeneration stand made with the digitizer Fastrak Polhemus.

**Figure 5 sensors-24-00109-f005:**
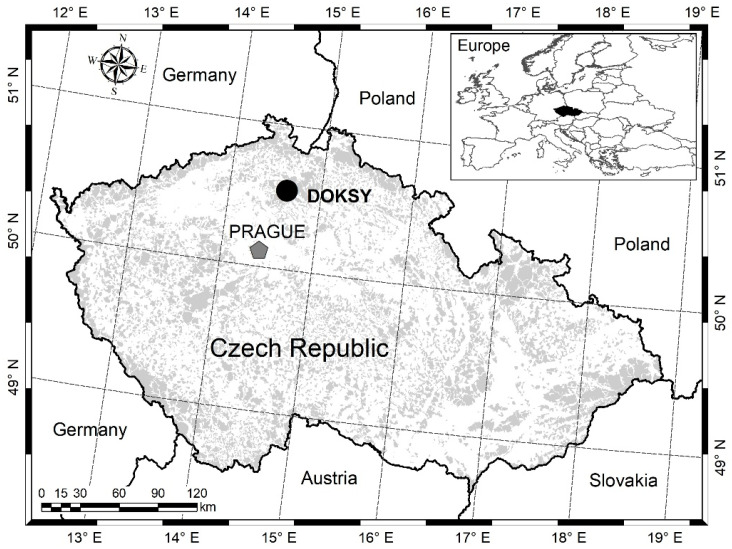
Location of the study area. The black dot represents the interest area in Doksy.

**Figure 6 sensors-24-00109-f006:**
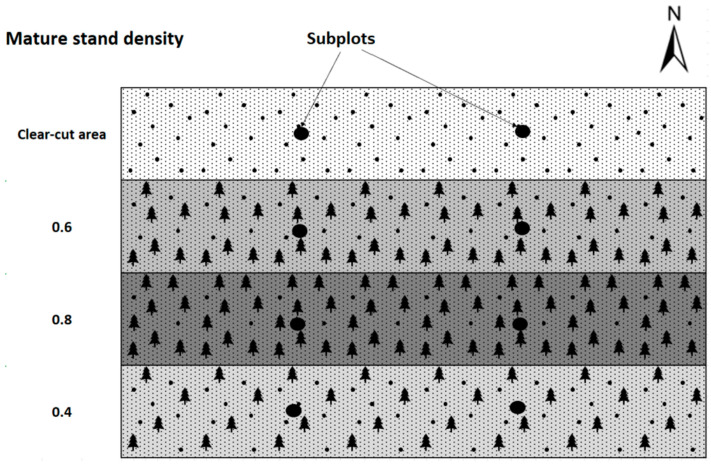
Experimental design, including the location of the subplots for natural regeneration sampling (black circles).

**Figure 7 sensors-24-00109-f007:**
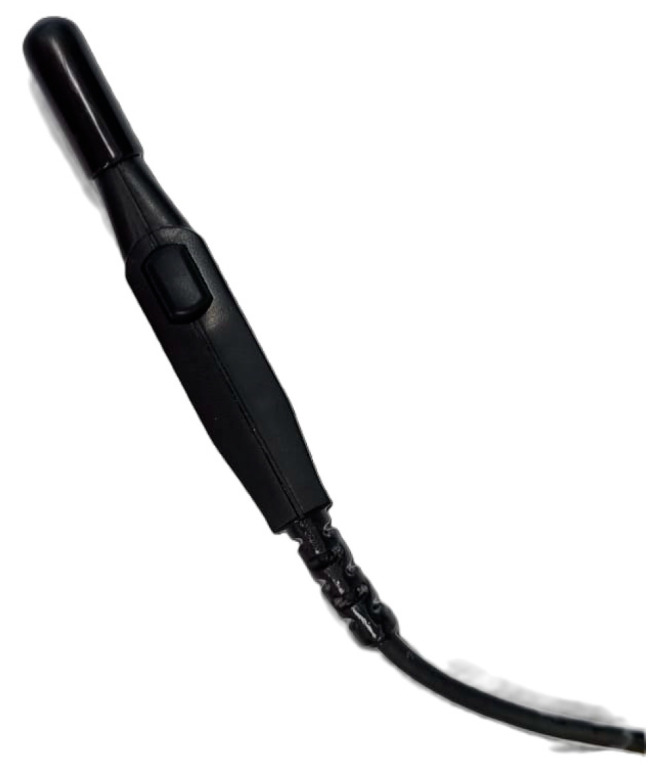
Stylus is a pen used to record the coordinates of an object in a given space.

**Figure 8 sensors-24-00109-f008:**
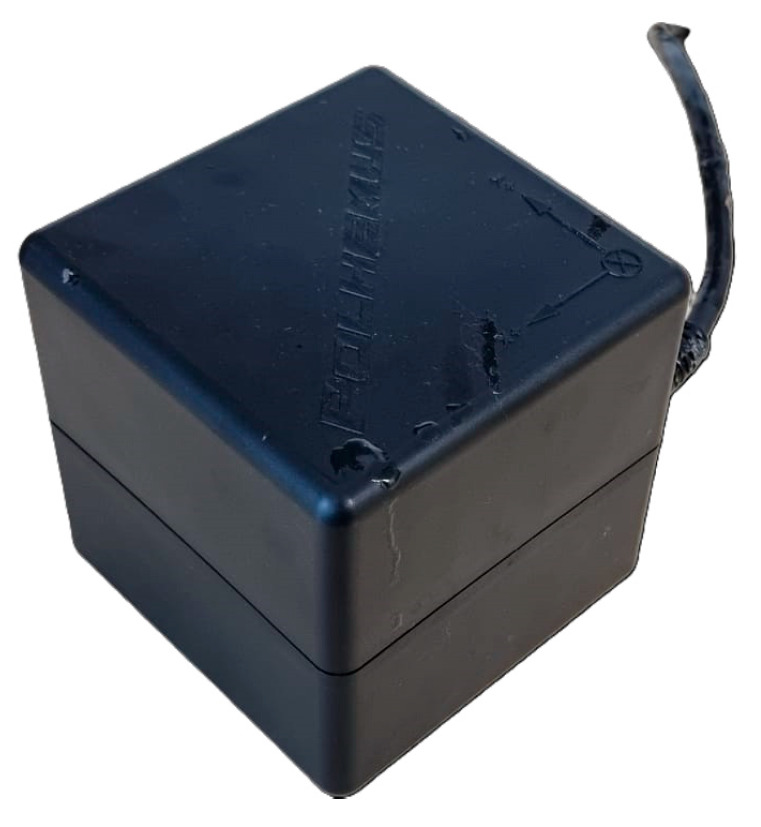
Generator of electromagnetic field.

**Figure 9 sensors-24-00109-f009:**
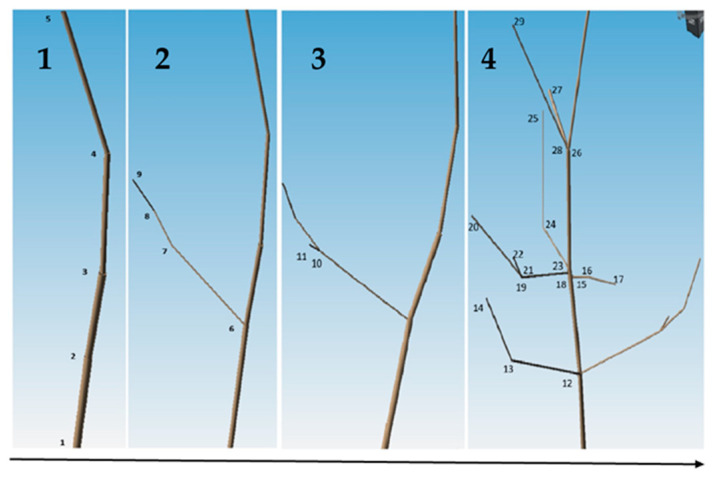
Graphical representation of the continuity of the process of 3D modeling of the aboveground plant part of Scots pine individuals. Figure (**1**–**4**) shows the sequential 3D modelling of Scots pine with the time sequence shown by the arrow below the figure. Numbers 1–29 are the places where a point is measured.

**Figure 10 sensors-24-00109-f010:**
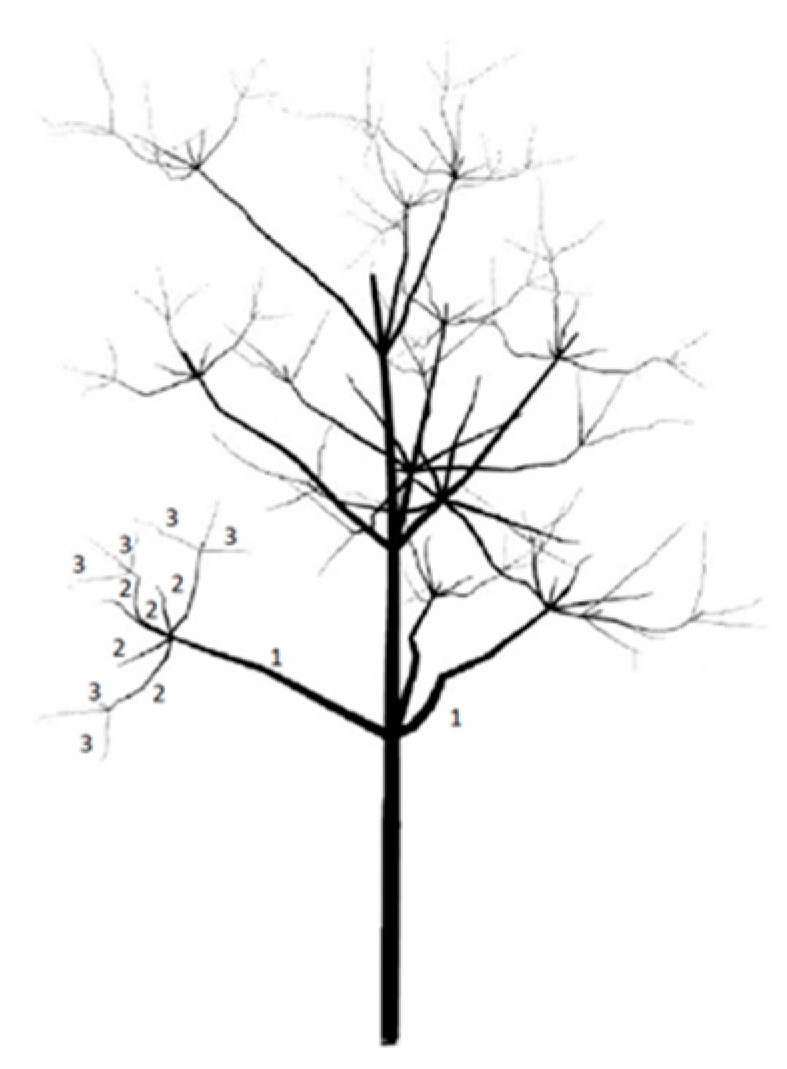
Description of the measured parts of the tree: 1—first-order branches, 2—second-order branches, 3—third-order branches.

**Figure 11 sensors-24-00109-f011:**
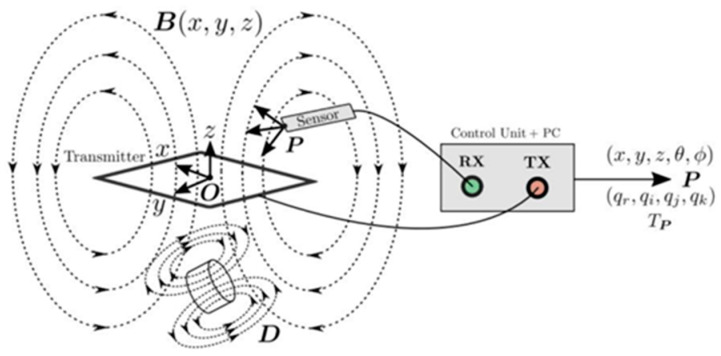
Universal operation of the magnetic tracking system. The control unit is connected to a transmitter board (TX) and a magnetic sensor (RX). The sensor is induced by the voltage from the time-varying magnetic field B of the transmitter. The controller measures this voltage and uses this value to determine the position of the sensor P relative to the origin O. The orientation component may be expressed as a vector, quaternion, or transformation. Distortions D in the TX plate region distort the magnetic field and cause errors in the determination of the sensor position [39].

**Figure 12 sensors-24-00109-f012:**
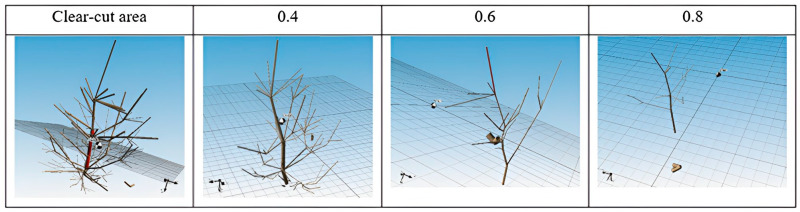
Sample of measured 3D models.

**Figure 13 sensors-24-00109-f013:**
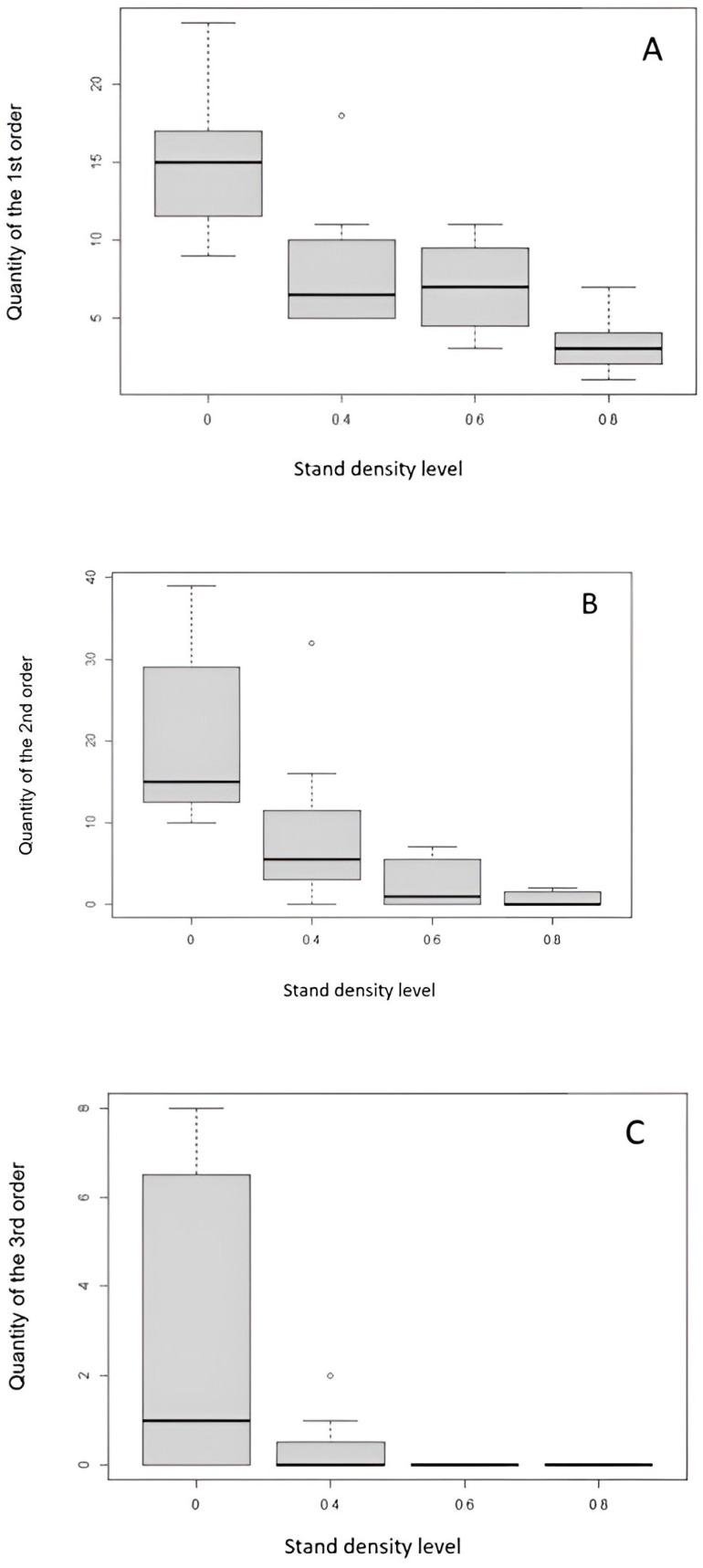
Amount of branching of occurrence of the first-(**A**), second-(**B**), and third-order branching (**C**) at different stand densities. The boxplots show the standard deviations (the gray bar), median (the black horizontal line), and minimum and maximum values.

**Figure 14 sensors-24-00109-f014:**
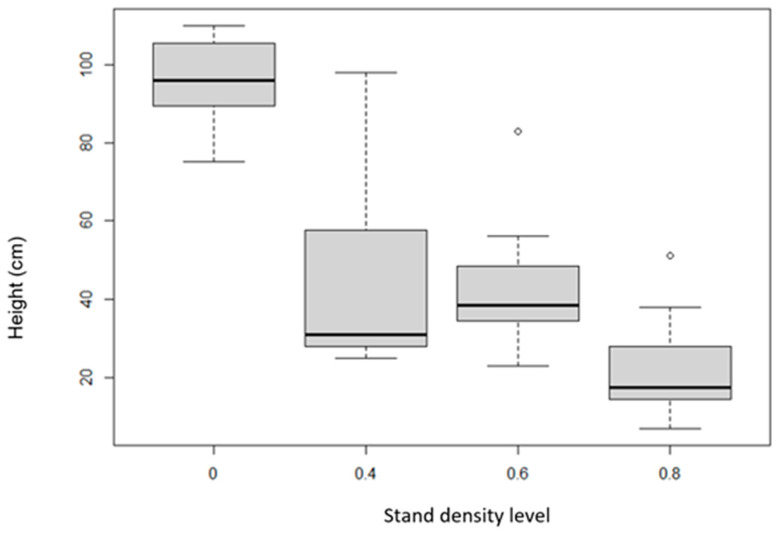
Amount of branching of the height of pine trees depending on stand density. The boxplot graph shows the standard deviations (the gray bar), median (the black horizontal line), minima, and maxima.

**Table 1 sensors-24-00109-t001:** The effect of stand density on the abundance of branches in each order.

Coefficients	Intercept	Stock Density (Estimate)	STD	Error from Value	Pr (>|z|)
Order 1	2.73680	−1.62223	0.20893	−7.765	8.19 × 10^−15^ ***
Order 2	3.11377	−3.28529	0.24351	−13.49	<2 × 10^−16^ ***
Order 3	1.1635	−6.4741	I.14	−4.721	2.35 × 10^−6^ ***

Notes: The results indicate that stand density has a significant effect on the abundance of branches in each order. The intercept represents abundance when stand density is zero. As the order increases, we observe larger changes in the estimated coefficients, which indicates a more complex relationship between stand density and number of orders. All low *p*-values (***) that are reported indicate statistical significance of this effect. Our analysis provides evidence that the stand density plays a key role in influencing the abundance of individual orders.

**Table 2 sensors-24-00109-t002:** Dependence of the height of individuals on the density of the parent stand.

Coefficients	Estimate	STD	Error from Value	Pr (>|z|)
Intercept	2.73680	0.09168	29.852	<2 × 10^−16^ ***
Stand density level	−1.6223	0.20893	−7.765	8.19 × 10^−15^ ***

Notes: Intercept—represents the expected tree height when the stand density is zero. Stand density level—expresses the expected change in the tree height increasing the stand density. ***—It indicates a higher level of significance, at a significance level of 0.01. The null hypothesis holds, and the probability is less than 1%.

**Table 3 sensors-24-00109-t003:** Dependence of the tree height on the total branch biomass in each order.

Coefficients	Estimate	STD	Error from Value	Pr (>|z|)
Order 1	0.087987	0.016221	5.424	1.10 × 10^−5^ ***
Order 2	0.013591	0.006651	2.043	0.05127
Order 3	−0.233641	0.080563	−2.900	0.00749 ***

“Order 1” is 0.087987, signifying a significant positive effect on individual height with a unit increase in first-order branch biomass (*p*-value 1.10 × 10^−5^ ***). Order 2: The coefficient for “Order 2” is 0.013591, with a *p*-value (0.05127) above the 0.05 significance level, suggesting potential insignificance, though some effect on height may still exist. Order 3: The coefficient for “Order 3” is −0.233641, indicating a significant negative effect of third-order branch biomass on individual height (*p*-value 0.00749 ***). ***—It indicates a higher level of significance, at a significance level of 0.01. The null hypothesis holds, and the probability is less than 1%.

## Data Availability

Data are available upon request to the authors.

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
