# Peer review of "Measuring the Canopy Architecture of Young Vegetation Using the Fastrak Polhemus 3D Digitizer"

_sensors, 2023, doi:10.3390/s24010109_

Round 1
Reviewer 1 Report
Comments and Suggestions for Authors
I trust the authors will find the following comments constructive.
The paper purports to present a method of measuring the canopy architecture of young vegetation using the Fastrack (sic) Polhemus 3D digitizer. I don't think the novelty of the research is sufficiently emphasised in the Abstract or the Conclusions.
The literature review is extensive though some of the literature is old and if it has not been superseded then a brief statement should be included to that effect. It also lacks context. I am unsure how the methodology and results of the research based on 15cm, 30cm and 90cm (see lines 312 and 313) specimens can be extrapolated to predict and estimate future growth and canopy fixation (see lines 28-29). This needs to be clarified.
The authors assert (line 382 and 383) that Nilsson states that Lidar underestimates tree height by 2.1 - 3.7 m (sic). Why is this relevant? The authors are dealing with (if I understand them correctly) with specimens ranging from 15cm to 90cm. This is analogous to comparing apples with oranges and lacks context as alluded to previously.
Some specifics that I hope the authors will find constructive;
Line 3, should Fastrack read Fastrak and so on throughout the paper?
Line 21, please clarify 'on-site 0.8'
Line 81, remove full stop and replace with (see Figure 2).
Line 92, (Figure 3)
Figure 3, some text is unclear.
Figure 6, increase the text size and clarity of the Figure. What units apply for Stand Density?
Line 196, precisely what material was obtained? Please spell it out.
Line 253, 'branch diameter parameter' what does this even mean, do you mean branch diameter?
Line 258, 'Other mathematical methods', what other mathematical methods, be specific.
Figure 13, this is a classic box and whisker plot and should be explained as such.
Line 72, 'Lidar must be flown either by plane or drone', this is irrelevant within the context of the paper, there are many handheld Lidar scanners.
Line 345, first mention of GroIMP in the paper, please elaborate and explain.
Table 1 needs a fuller analysis and discussion.
Comments on the Quality of English LanguagePlease be as clear and succinct as possible. It will commend your work to a wider readership. Sometimes it's better to be clear than clever only.
Reviewer 2 Report
Comments and Suggestions for Authors
The paper describes the application of a new technology for measuring the architecture of young trees. It is useful topic, but I suggest to make some changes in the manuscript.
· The description of Fastrack Polhemus is rather long (lines 87-116) but not enough clear. There are many details (e.g., about effect of metallic parts and coils vs stylus), but the clear description what namely it measures and how technically (what do you do with these styluses or coils) is missing. Later (lines 238-247) there is a clear description of the method. I suggest to shorten the first description. Simply saying, the method allows to measure 3D coordinates of manually selected points relatively to a certain starting point - did I understood well? If yes, it makes sense to write it in the Introduction. Also, you should mention that the method is mainly applicable to small trees, as you need to reach manually every measuring point (if I understood well). So, for full grown tree you will need ladder or scaffold and the measurement will take a lot of time.
· I also suggest to shorten dramatically the discussion about tree height measurements (p. 4.2), (1) because the negative dependence of understory height on overstory density (especially for such a heliophyte tree as Skots pine) is well known fact and (2) because you do not need such a sophisticated equipment to measure tree heights – you can use tape-measure or geodetic rod to measure much higher number of trees during the same time. In contrast, if you have full data of sample trees architecture, you can add more information just on this topic (e.g., number of 1st order branches in one whorl, branch lengths, maybe to add pictures with examples of tree structures (like Fig. 12, but showing concrete trees, not scheme) etc.), showing the usefulness of applying this technology.
· It makes sense to notice more clearly if the main aim of the crown structure analysis is the providing necessary quality of industrial wood (industrial aim) or the evaluation of regrowth survival (ecological aim).
· Did you measure only branches or also roots (as trees were harvested with roots, L. 190)? If not, why to harvest trees with roots?
· In the description of regression models uncommon names are applied (e.g., value of the statistical linear model, error from value), so it is not clear what is mentioned.
· Some other notes are in the attachment

English, to my knowledge, is OK, some small notes are in the attachment
Reviewer 3 Report
Comments and Suggestions for Authors
General Assessment
Research Field: The study falls within the domain of forest management and 3D digitization for measuring canopy architecture.
Key Objective: To demonstrate the use of the Fastrack Polhemus 3D digitizer in assessing the structure of canopy architecture, particularly in young Scots pine, and how this technology might provide better insights compared to traditional methods like Lidar.
Methodology
Approach: The paper describes using the Fastrack Polhemus magnetic digitizer to create 3D models of young Scots pine trees. It compares this method against others like Lidar and highlights its precision and ability to capture intricate canopy details.
Data Collection and Analysis: The study involved collecting data from young pine trees in different stand densities and analyzing the branching patterns and tree heights. Statistical methods and regression models were used to interpret the data.
Results
Findings: The study found a correlation between stand density and the number of branches in Scots pine, as well as an inverse relationship between branching intensity and tree height. The Fastrack Polhemus digitizer proved efficient in capturing detailed 3D models, offering advantages over Lidar in capturing overlapping branches and internal crown structures.
Statistical Analysis: The statistical methods are sound, and the results are well-presented with clear graphical illustrations.
Discussion
Interpretation of Results: The authors discuss how the findings can be applied in forest management, specifically in optimizing canopy architecture for better growth and carbon fixation in Scots pine.
Comparison with Previous Studies: The study contextualizes its findings within the broader research landscape, comparing its results with existing literature and highlighting the advantages of its chosen methodology.
Strengths
Innovative Methodology: The use of the Fastrack Polhemus digitizer for detailed 3D modeling is a significant contribution to forestry research.
Comprehensive Data Analysis: The study provides a thorough analysis of the collected data, offering valuable insights into the canopy architecture of Scots pine.
Weaknesses and Suggestions
Sample Size and Diversity: The study might benefit from a larger and more diverse sample size to strengthen the generalizability of the findings.
Comparative Analysis: Further comparison with other tree species and different environmental conditions could provide a broader perspective on the applicability of the technology.
Long-Term Impact: Investigating the long-term implications of canopy architecture on tree growth and forest ecology would be beneficial.
Overall Evaluation
The paper is well-structured, with a clear methodology, comprehensive data analysis, and a thoughtful discussion of the findings. It makes a significant contribution to the field of forestry research and offers valuable insights into the use of 3D digitization technologies in studying canopy architecture. The study's limitations are minor and do not detract from its overall quality and relevance.
Round 2
Reviewer 1 Report
Comments and Suggestions for Authors
Please see below
Comments on the Quality of English LanguageThe paper would benefit from another thorough read by the authors in order to remove occasional minor spelling, grammar and formatting issues.
Author Response
1) The paper would benefit from another thorough read by the authors in order to remove occasional minor spelling, grammar and formatting issues.
Dear reviewer
we passed through the text and also asked an additional 2 authors to
verify the meaning of phrases and also to find mistakes. We put attention that the meaning remains the same with all the small changes. Thank you again for your constructive comments.